# Physicomechanical Properties of Rice Husk/Coco Peat Reinforced Acrylonitrile Butadiene Styrene Blend Composites

**DOI:** 10.3390/polym13071171

**Published:** 2021-04-06

**Authors:** Nurul Haziatul Ain Norhasnan, Mohamad Zaki Hassan, Ariff Farhan Mohd Nor, S. A. Zaki, Rozzeta Dolah, Khairur Rijal Jamaludin, Sa’ardin Abdul Aziz

**Affiliations:** 1Razak Faculty of Technology and Informatics, Universiti Teknologi Malaysia, Jalan Sultan Yahya Petra, Kuala Lumpur 54100, Malaysia; ain.kl@utm.my (N.H.A.N.); rozzeta.kl@utm.my (R.D.); khairur.kl@utm.my (K.R.J.); saa.kl@utm.my (S.A.A.); 2Malaysia-Japan International Institute of Technology, Universiti Teknologi Malaysia, Jalan Sultan Yahya Petra, Kuala Lumpur 54100, Malaysia; ariffmdnor@yahoo.com (A.F.M.N.); sheikh.kl@utm.my (S.A.Z.)

**Keywords:** biocomposites, blend, recycle composites, biodegradable

## Abstract

Utilizing agro-waste material such as rice husk (RH) and coco peat (CP) reinforced with thermoplastic resin to produce low-cost green composites is a fascinating discovery. In this study, the effectiveness of these blended biocomposites was evaluated for their physical, mechanical, and thermal properties. Initially, the samples were fabricated by using a combination of melt blend internal mixer and injection molding techniques. Increasing in RH content increased the coupons density. However, it reduced the water vapor kinetics sorption of the biocomposite. Moisture absorption studies disclosed that water uptake was significantly increased with the increase of coco peat (CP) filler. It showed that the mechanical properties, including tensile modulus, flexural modulus, and impact strength of the 15% RH—5% CP reinforced acrylonitrile-butadiene-styrene (ABS), gave the highest value. Results also revealed that all RH/CP filled composites exhibited a brittle fracture manner. Observation on the tensile morphology surfaces by using a scanning electron microscope (SEM) affirmed the above finding to be satisfactory. Therefore, it can be concluded that blend-agriculture waste reinforced ABS biocomposite can be exploited as a biodegradable material for short life engineering application where good mechanical and thermal properties are paramount.

## 1. Introduction

Researchers have recently been looking for high-performance materials based on their lightweight capability and assembly features toward the end product’s lowest cost. Not all available products in the market fulfill the designers’ need. Therefore, the researchers move toward the advanced material called composite materials. The composite material is a combination of two or more constituent materials with different physical and chemical properties. When combined, they produce unique properties compared to individual material. In general, composite materials offer excellent weight to strength ratio, thermal, moisture uptake, and wear properties. Engineering composite is a combination composite materials harder and stronger phase, which is called reinforcement material, and the stiff continuous segment is termed matrices. The matrix can be either thermosetting or thermoplastics, while the reinforcement material could be metals, ceramics, or fibers. Previously, synthetic fibers, including glass [1,2], boron, carbon, and Kevlar, were famously used for reinforcement. However, due to their high cost and non-biodegradable property, scientists and technologies shifted to full usage of natural fibers and agro-waste materials. Recently, vegetable fibers and crop residues like banana [3], kenaf [4], and bamboo [5], coco peat (CP), and rice husk (RH), were evaluated to reinforce polymer composite materials and were the best candidates to replace synthetic fibers. Besides being recyclable, the plant fibers haves offer many advantages, such as low cost, low density, and abundant availability. Due to ecological concern and new rules and regulations, the development of a new biocomposite that consists of thermoplastic polymer and agro-waste fiber as reinforcement is mandatory. Consequently, this study will reduce the carbon cycle, decrease environmental impact, and thus produce a greener product.

Generally, rice (*Oryza sativa*) is one of the essential agricultures product in this world. In 2017, the global annual production of 670 million tons of paddy was harvested from Asia, America, Africa, and Europe [6]. Approximately 20% of the paddy were estimated to be rice husks. In Malaysia, a total of 840 thousand tons of RH are produced annually [7]. The utilization of RH in composite structures is recommended in many engineering applications due to its abrasive nature, low cost, lightweight, renewability, biodegradability, universally available, and weathering resistant. It has been reported that the incorporation of RH into thermoplastic matrices enhances the mechanical [8], flammability [9], and thermal stability [10] properties of biocomposites. Chen et al. [8] reported that the RH filler sample improved up to 58% tensile modulus than neat polymer. Moreover, with the addition of RH in the eco-composite structure, the flammability heat release rates were significantly dropped by approximately 39%, which can be attributed to the presence of silica in RH [9]. Additionally, the degradation rate values for RH reinforced polymer were shifted to a higher temperature, indicating improved thermal stability of the biocomposite as compared to recycled polymer [10]. Many studies also mentioned that RH-filled composite material has a low moisture absorption kinetic and good dimension stability than other natural fiber composites [11]. High moisture absorption capability of the biocomposite results in weak interfacial bonding which reduces microbial resistance, is easy to buckle under compressive loading, and contributes to deterioration of mechanical properties. Therefore, these moisture absorption characteristics are essential and critical factors to be evaluated for potential use in outdoor application. Several research works have assessed the water absorption behavior of RH composites upon immersion in distilled water.

In addition, coco peat is a spongy particle and by-product waste of coconut shell. It is a rich source of lignocellulose consisting of lignin, cellulose, and hemicellulose. Traditionally, this coconut waste is abundantly available and also widely used as soilless potting mix media in agriculture. A coconut mostly contains approximately 100 g of coco peat with a mixture of coarse-to-fine cork particles (83–95% in total) and fibers. Currently, about eight million tons of coco peat are being produced from coconut husks in the world each year. In Malaysia, approximately 5280 kg per hectare of coconut waste, mainly coconut husk, is obtained, but most have not yet been processed and fully utilized. Coco peat is highly plausible with adequate water capacity storage, high water retention, and eco-friendly. However, it has a moderate mechanical property. Borawski [12] stated that unconventional materials like coconut waste are an excellent alternative for many automotive applications, such as brake pads. Besides, hybrid coco peat composites also offer higher wear resistance to manufacture clutch plates lining and superabsorbent capability for desiccant evaporating cooling systems [13].

Based on past findings, even though extensive research has been done to explore composites’ physical and mechanical properties, very few involved blend biocomposites. The claims of excellent RH composites on mechanical and especially tensile properties and low moisture absorption mentioned before, together with abundantly available but possess moderate mechanical properties of CP fiber, became the biocomposites’ filler selection to run the investigation. Indeed, the aim of this study was to explore the synergistic effects of the blended RH and CP filler reinforced ABS composite on the physical and mechanical properties that has not been conducted previously. However, only a handful of publications raised concerns about the moisture characteristics of the blended composite, especially with coco peat and acrylonitrile butadiene styrene (ABS). Therefore, the moisture absorption, tensile behaviors, flexural test, and impact resistant of these biodegradable composites were tested. Besides, the composites were tested according to ASTM standards, and the surface morphological were finally evaluated experimentally. This investigation direction is beneficial for producing green waste composite materials with a synergistic combination of cost and performance. This topic remains unexplored at the time of writing, and the need to investigate this area of matter is crucial to fill the knowledge gap.

## 2. Materials and Methods

### 2.1. Materials

Acrylonitrile Butadiene Styrene (ABS) was supplied by Muleh Zaman Enterprise (Gombak, Selangor, Malaysia) in pallet form and was used as the polymer matrix. The average pellet size was 4 mm. Rice husk (RH) particle and cocoa peat (CP) were purchased from Innovative Pultrusion Sdn. Bhd. (Seremban, Negeri Sembilan, Malaysia). RH has an average density of 0.22 g/cm^3^ specifically from *Oryza sativa* species and CP has an average density of 0.10 g/cm^3^ solely from *Cocos nucifera* species.

### 2.2. Sample Fabrication

Initially, RH particle sized ±10 µm, CP particle sized ±0.25 mm, and thermoplastic ABS in pellet form were dried in an electric air circulated oven (CMH Ltd., Lancing, UK) at 80 °C for 48 h. Five sets of RH/CP (20/0, 5/15, 10/10, 15/5, and 0/20) wt.% reinforced thermoplastic ABS were fabricated as tabulated in Table 1. The RH/CP blend composites were prepared via the melt blend internal mixer (Brabender, Duisburg, Germany) at an optimum processing temperature of 190 °C and a rotating speed of 40 rpm. The composites were then oven-dried (CMH Ltd., Lancing, UK) at 80 °C for 2 h [14], followed by chopping (Cheso N3, Loyang Way, Singapore) the composite to form pallets. The composite granules were then fed to a screw-type injection molding (Engel Gmbh, Schwertberg, Austria). It was further mixed, heated, and then extruded through three plate mold dies to create uniform distribution tensile, three-point bending, and impact test samples.

### 2.3. Physical Properties of RH/CP Reinforced ABS Blend Composites

The density of biocomposites was evaluated according to ASTM D4018. The thickness and weight of the developed blend composites of RH/CP were recorded for tests. The samples were weighed to the nearest 0.001 g by using a close chamber EMS 300-3 precision balance (Kern and Sohn, Balingen, Germany).

Water absorption characteristics were conducted following the ASTM D570. The sample was initially dried in a circulation oven (CMH Ltd., Lancing, UK) at 80 °C for 2 h before the composites’ weight was obtained. In this study, five replicates of composite specimens with the dimension of 20 mm × 20 mm × 3 mm were immersed in distilled water at room temperature of 25 °C. The coupons were taken out from the moist condition, and all exposed surfaces were dried by using a microfiber cloth (Spontex, Worcester, Worcestershire). The moisture uptake data were recorded by using a weight balance EMS 300-3 (Kern and Sohn, Balingen, Germany) regularly at every 2 h of water immersion. The moisture absorption characteristic was evaluated by using the following equation:(1)$$\%M=\frac{W_{t}-W_{0}}{W_{0}}\times 100$$
where, *W_t_* is the sample’s weight at a recorded immersion time, and *W*_0_ is weight of the dried sample. The kinetic and diffusion mechanism was evaluated based on Fickian’s theory. Its relation is:(2)$$\log\left[\frac{M_{t}}{M_{\infty }}\right]=\log\left(k\right)+n\log\left(t\right)$$
where *M_t_* and *M*_∞_ are water absorption at time, *t* and saturation point, respectively. *k* and *n* are constants.

The ability of moisture to penetrate the composite’s molecule, also known as diffusion coefficient, *D* is a primary parameter of Fick’s model [15]. The *D* value is obtained from initial linear portion of the moisture absorption percentage versus the square root of the time curve. The one-dimensional diffusion coefficient, *D*, can be determined from the following equation:(3)$$D=\pi {\left[\frac{h}{4M_{\infty }}\right]}^{2}{\left[\frac{M_{2}-M_{1}}{\sqrt{t_{2}}-\sqrt{t_{1}}}\right]}^{2}$$
where *h* is the plate thickness, *M*_2_, *M*_1_ are the moisture content at time *t*_1_ and *t*_2_, respectively.

### 2.4. Tensile Test

Tensile properties of the RH/CP reinforced ABS blend composites were performed according to the ASTM D618 test standards. The samples were fabricated into flat dog-bone shaped to accommodate the calibrated universal testing machine (AG-X plus 50 kN, Shimadzu, Kyoto, Japan). The speed of the tensile testing was fixed at 1 mm/min. The tests were performed on five samples, and the average reading was taken as the final result.

### 2.5. Flexural Test

Flexural properties of the fabricated RH-CP/ABS blend composite were evaluated according to the ASTM D790-03 (3-point bending) standard. The testing was executed by using a universal testing machine (AG-X plus 50 kN, Shimadzu, Kyoto, Japan) with a span to depth ratio of 16:1. The flexural testing speed was fixed at 1 mm/min by using a 50 kN of the load cell.

### 2.6. Impact Test

In this study, the unnotch Charpy impact test was used to measure the impact characteristic of the fabricated RH-CP reinforced ABS blend composite. Samples with a dimension of 65 mm × 12.7 mm × 3.2 mm were evaluated by using a pendulum impact tester (Zwick 5113, Ulm, Germany). The pendulum was released at an energy capacity of 4 J and release angle of 160°. This experimental procedure was conducted according to ASTM D256 test standards. An average of five samples relative to the RH-CP reinforced ABS blend composites’ impact strength was evaluated.

### 2.7. Composite Characterization

The morphological investigations of RH-CP reinforced ABS blend composite were examined by using a field emission scanning electron microscope (FESEM) (JEOL, JSM-7800F, Tokyo, Japan). The composite samples were then analyzed under a magnification of 500× at an accelerating voltage of 2 kV. Prior to the evaluation, the samples were initially coated with platinum by using a fine auto coater (JEOL, JEC-3000FC, Tokyo, Japan).

## 3. Results and Discussion

### 3.1. Density of Blend Composites

The measured density values of the RH/CP reinforced thermoplastic ABS composites are illustrated in Figure 1. This composite density was subjected to various particle loading of RH/CP (0/20, 5/15, 10/10, 15/5, and 20/0) in a weight fraction. It was shown that by increasing the rice husk to coco peat in blend composites, the density also increases. The highest density result was obtained from RH20/CP0 as compared to other biocomposite configurations. This result was mainly ascribed to the higher density of RH to the CP filler. Similarly, Hemnath et al. [16] reported that by increasing the RH content the void content decreased due to smaller particle size in blend composites. Higher RH loading reduces the pores content resulting in tighter and pack composites. This void percentage of the composite can be controlled by adjusting the reinforcement amount and molding parameters during fabrication. These lead to improved mechanical properties, sound absorption capabilities, and thermal insulation of the structures.

### 3.2. Moisture Absorption Study

The moisture absorption trends of the RH/CP (0/20, 5/15, 10/10, 15/5, and 20/0) reinforced ABS are illustrated in Figure 2a. The plots were deducted from an average value of three specimens. For immersion time lower than 20 h, the moisture absorption (%) rate steadily increased with increasing coco peat content. The maximum moisture absorption is from the RH0/CP20 wt.% of coco peat composite composition. It can be suggested that the hydrophilicity of the coco peat particle was higher than the rice husk filler. Deo et al. [17] reported that weak fiber-matrix adhesion and void content influenced the natural fiber composites’ moisture uptake. The high content of coco peat filler increased free hydroxyls (OH) groups in cellulose. These OH groups enhanced the contact with moisture and form several hydrogen bonding, resulting in weight gain in the blend composites. In general, moisture uptake increases with immersion time. However, it remained as a constant plateau after 40 h, as shown in Figure 2a. The time to reach a saturation condition was nearly similar for all tested samples. The extent of blend composites in a humid environment promotes an increase in the swelling dimension and low-stress transfer between particle and matrix, which corresponds to the reduction in mechanical properties [18].

Figure 2b presents a typical log (*M_t_*/*M*_∞_) versus log (*t*) for the RH/CP reinforced ABS composites. The swelling transport exponent ‘*n*’ and characteristic constant (*k*) of the moisture absorption values were determined from the slopes and intercepts of these plots, respectively. The results indicated that the diffusion exponent for the composite samples lies between 0.56 and 0.59. This can be explained that the RH/CP reinforced ABS was in the range of Fick’s model. Similar findings were reported by Razavi et al. [19] and Awasthi et al. [20]. The *n* values were between 0.5 and 0.7 when the Fickian diffusion property was applied in the studies to determine the rice husk reinforced thermoplastic composites’ moisture absorption characteristic. It was claimed that these biocomposites were in an intermediate diffusion process between the penetrant mobility cases I and II. However, Guloglu et al. [21] mentioned that most composites structures were in a range of abnormal water diffusion characteristics. This typical non-Fickian moisture absorption behavior of the thermoplastic-based composites was also reported by Melo et al. [22] and Aziz et al. [23] when the samples were continually exposed to a wet environment at a lower absorption rate and prolonged period. 

Moreover, the blend composites *k* value increased with increasing coco peat content, as tabulated in Table 2. It indicated that the higher the coco peat filler in composites, the higher the biocomposites kinetic water absorption characteristic. A higher value of *k* explained that the fastest blend composite diffusion time is needed to reach a saturation condition. Findings were also aligned with the result, as illustrated in Figure 2a. Similar results were reported by Guna et al. [24]. It was mentioned a decreased water resistance with the increase up to 40 wt.% of coconut filler content of the hybrid composites, resulting in insufficient resin to impregnate and wet out the fillers. In this study, the *D* values of the biocomposites also increased with increase in coco peat content. At lower rice husk composition, the water dispersion mobility rate in a composite capillary was low, resulting in lower *D* values. These *D* values were in a range reported by Chen et al. [8]. It demonstrated that the pack of hybrid arrangement in rice husk filler would reduce the moisture kinetic absorption due to narrow gaps and voids formation in the biocomposites. A similar finding was reported by Nanthakumar et al. [25], which suggested that an improvement of filler-matrix adhesion by using a surface modification technique could reduce water molecules to diffuse and penetrate the composite structures. Besides, an extra information on the blend composites’ glass transition temperature helped to predict the plasticizing effect of solvent on the polymer [26,27].

### 3.3. Tensile Properties

Five types of RH/CP fiber weight percentages were used in blend composites. RH and CP fiber used as the fiber content in the composites were fixed at 20 wt.% while the epoxy resin matrix was fixed at 80 wt.%. The tensile study was conducted and showed that the tensile properties were affected by the variation in blend compositions. The tensile properties are shown in Figure 3. It was demonstrated that RH15/CP5 composition showed the highest tensile strength among the blend composites. Tensile strength enhancements of 18.4% and 17.8% were reported for RH15/CP5 blend composites were achieved as compared to its monofiber composites for RH0/CP20 (coco peat monofiber composites) and RH20/CP0 (rice husk monofiber composites), respectively. This phenomenon was due to the blending ability to practically overcome the traditional low strength disadvantages of single type natural fiber-reinforced composites [28,29]. Practically, the RH particle size was much smaller than that of CP. Consequently, the surface area available for wetting by the polymeric matrix was higher with higher RH filler loadings. Under quasi-static tensile loading, stress transfer from matrix to filler may be more efficient in the RH dominant composites, resulting in higher mechanical properties [30]. Moreover, the tensile strength also revealed an increment as the RH content increases. Simultaneously, the CP content was decreased, up to 15 wt.% RH and down to 5 wt.% CP, respectively, on the composites. The composite properties mainly depended on the mechanical behaviors of the individual reinforcing fibers [31]. RH was reported as a kind of natural fiber with higher stiffness in tensile properties as compared to CP fiber [32]. This event was explained by the tensile strength increment as the RH loading extends and not the opposition’s extension of CP loading. However, the tensile strength decreased with further addition of 20 wt.% RH content, by as much as 15.1%. Poor interfacial bonding with the evidence from the micrograph of a fractured specimen in Figure 8 was the reason for the tensile deterioration. Aridi et al. [33] studied the mechanical properties of RH polypropylene composites, and the results showed that as the RH composition increased to 55 wt.%, the strength decreased. It was concluded that the weak bonding between the hydrophilic filler and the hydrophobic matrix polymer obstructed the stress propagation, and thus caused the tensile strength to fall when the filler loading increased. Besides, poor dispersion caused agglomeration of the fillers and acted as stress concentration points, which led to composite failure [34]. This caused a decrease in tensile strength.

Nevertheless, the addition of RH fiber loading instead of CP fiber loading proportionally increased the tensile modulus. It was deduced that the tensile modulus was affected by the blending of RH and CP fiber as the reinforcement filler. The highest enhancement of 16% on tensile modulus was detected from RH20/CP0 in contrast with RH0/CP20. This finding remarked that the composites’ stiffness came from the stiffness of CP, particularly on top of the ABS itself. Altogether, it was observed that the RH20/CP0 blend composite delivered the maximum Young’s modulus, whereas the lowest was delivered by RH0/CP20 filler for the composites. Figure 4 demonstrates the tensile stress-strain curves for RH20/CP0, RH15/CP5, RH10/CP10, RH5/CP15, and RH0/CP20 blend composites from the tensile test. The curve deduced that the 15 wt.% RH and 5 wt.% CP resulted in the highest value in tensile stress. Meanwhile, the tensile strain’s highest extent was achieved by the 20 wt.% CP blend composites. The tensile strain recorded that the composites’ strain increased as the amount of CP fiber filler increased. The monofiber CP composites showed an improvement of 15.5% strain as compared to the monofiber RH composites. Therefore, the CP fiber upsurged the ductility behavior of the blend composites better than RH fiber.

### 3.4. Flexural Properties

In order to further understand the mechanical properties of the blend composites, flexural tests were conducted. Figure 5 presents the flexural strength and flexural modulus results obtained from the tests. It was apparent from this figure that the trend for this test was similar to the tensile test. The finding provided evidence that the flexural strength increased as the RH fiber filler increased, up to 15 wt.%, then decreased after the addition of 20 wt.% RH fiber to the blend composites. Further analysis showed that the RH15/CP5 blend composite revealed the highest value of flexural strength in comparison with its RH and CP monofiber composites, with an improvement of 9.0% and 2.3%, respectively. The flexural strength improvement was not quite as significant as the tensile strength improvement, but still considerably affecting the composites. The lowest flexural strength was RH0/CP20, likewise the tensile strength discussed in the previous section. There were several possible explanations for this result. A possible reason for this might be that a weak link may exist between the interfacial bond between hydrophilic CP fiber and hydrophobic resin [35]. 

In Figure 5, a clear trend is shown on the increasing flexural modulus as the percentage of introducing RH filler into the composites becomes larger. The optimum flexural modulus was found on the RH20/CP0 at 5.51 GPa, followed by RH15/CP5, RH10/CP10, RH5/CP15, and RH0/CP20. As mentioned in the literature review, no data was found on the blending between RH fiber and CP fiber into a composite for comparison. On the other hand, Figure 6 illustrates the typical flexural stress-strain curves of the blend composites. Contrary to expectations, the result did not show any significant difference between the flexural strain enhancement with the variation of RH and CP fibers filler content. The insignificant difference in the flexural strain could be attributed to the elasticity among the composite, which was almost similar for each of them.

### 3.5. Impact Response Behavior

The impact response of the rice husk and coco peat filler blend ABS composites is summarized in Figure 7. It corresponds to the combination of RH/CP configuration. The incorporation of RH in the CP-ABS varied the impact behavior of the blend composite. Referring to the plot, the maximum impact energy and impact strength were obtained at RH15/CP5 wt.%. The value of the impact properties significantly increased with the increase of RH from 0 wt.% to 15 wt.%. Further increase in the proportion of biomass in the biocomposite reduced the impact strength due to delamination and insufficient matrix [36] to proper adhesion, and resulting in decreased strength. The overall impact properties of the blend composite can be improved by chemical coupling and surface modification. However, high modification can increase production cost and reduces the agriculture composite residues that are value-added.

### 3.6. Fracture Surface Micrograph

Figure 8 demonstrates the differences of fracture surface by using SEM for the composite containing RH15/CP15 and RH0/CP20 filler content. Filler fracture and matrix delamination are predominated in the RH15/CP15 specimen’s morphological surface, as illustrated in Figure 8a. Matrix cracking and the presence of void on the interface was also observed. It can be suggested that the stress was well scattered between the filler and thermoplastic matrix due to the clean braking surface. It was also indicated that this composite failed in the brittle mode failure. This means that an excellent interface bonding between filler and matrix was formed, resulting in high tensile strength property. Matrix cracking and fiber breakage were observed as in Figure 8b. As shown from this figure, more cavities were formed in this blend composite resulted from fiber pullout and air trap. It also demonstrated fracture damage occurred at the coco peat filler surface and which may suggest a flaw in these composites. It can lead to stress concentration at this weak point, and it was attributed to the lower mechanical properties. Arslan et al. [37] mentioned that the use of silane coupling agents such as (3-aminopropyl) triethoxysilane (AP) and 3-(trimethoxysilyl) propyl methacrylate (MA) could enhance the tensile properties of the biocomposite due to covalent bond formation between the amino group of coupling agent and the nitrile group of styrene-acrylonitrile (SAN) matrix. However, extra production cost would occur during fabrication.

## 4. Conclusions

The blend RH/CP reinforced with ABS was successfully prepared by using two-step processes: melt blend mixer and injection molding. The physicomechanical properties of these biocomposites were then characterized. It was found that the density of the composites increased with increase in RH filler. In contrast, the composite’s kinetic water absorption and moisture saturation increased steadily with CP content and obeyed Fick’s model. It was also found that the incorporation of RH and CP into the ABS matrix offered better performance in tensile, flexural, and impact strengths. The synergistic effect study clearly shows that the incorporation of RH and CP into the ABS matrix performs better on fiber’s interfacial bonding, as compared to the performances of individual composites’ components, and thereby enhances the mechanical performances of the overall system.

The highest tensile properties and Young modulus were recorded at the composition of RH15/CP5 wt.%, resulting in lower elongation at break as compared to RH0/CP20 wt.% blend composite. As expected, the incorporation of 15 wt.% RH improved the maximum flexural and modulus properties value more than those blend composites based on the ABS matrix. In addition, this combination was attributed to the highest value of impact strength. The fracture surface morphology for the blend composite dominated by matrix cracking and filler fracture explained the stress was well propagated between filler and matrix thermoplastic. It proved that the excellent adhesion interfaces bonding of the biocomposites resulted in maximum mechanical properties. Therefore, it was interesting to combine these natural waste reinforced ABS composites for a short life engineering application, where the physicomechanical properties of biocomposites are of paramount importance.

## Figures and Tables

**Figure 1 polymers-13-01171-f001:**
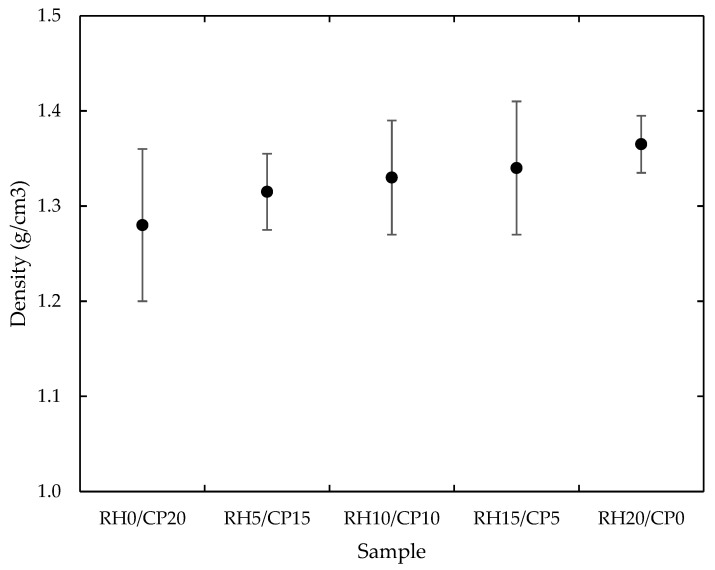
Typical density of rice husk (RH)/coco peat (CP) reinforced acrylonitrile-butadiene-styrene (ABS) blend composites.

**Figure 2 polymers-13-01171-f002:**
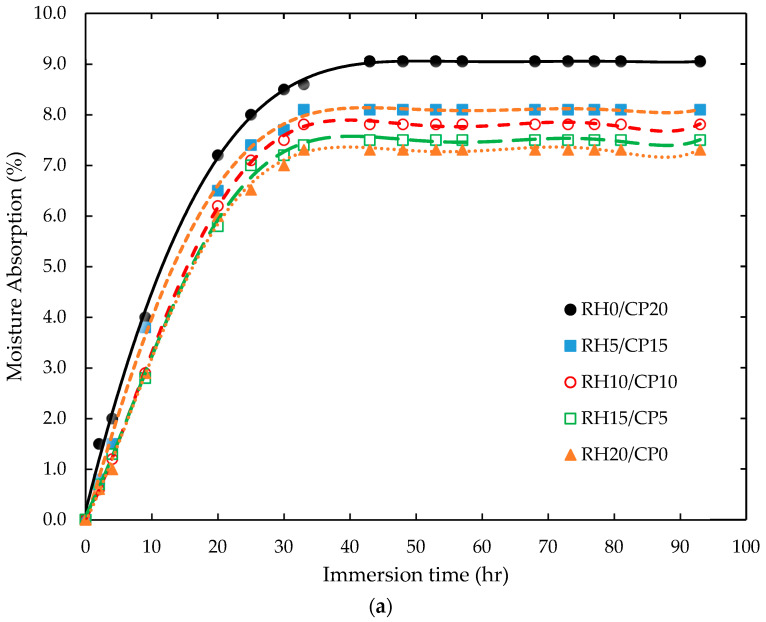
Typical (**a**) moisture absorption characteristics and (**b**) plot of log *M_t_*/*M*_∞_ versus log time of blend RH/CP reinforced ABS composites.

**Figure 3 polymers-13-01171-f003:**
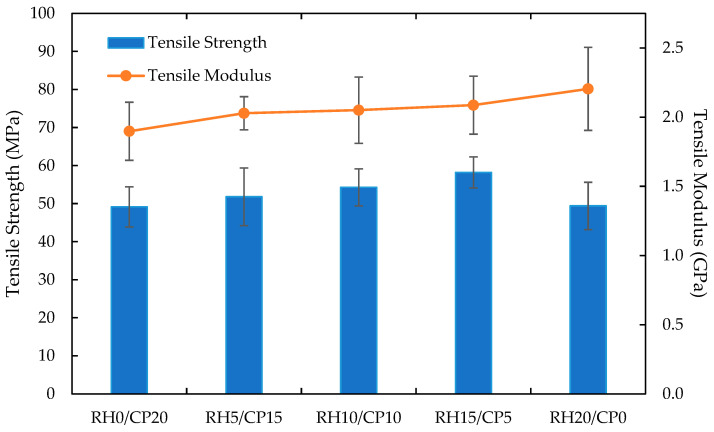
Tensile properties of RH/CP reinforced ABS blend composites.

**Figure 4 polymers-13-01171-f004:**
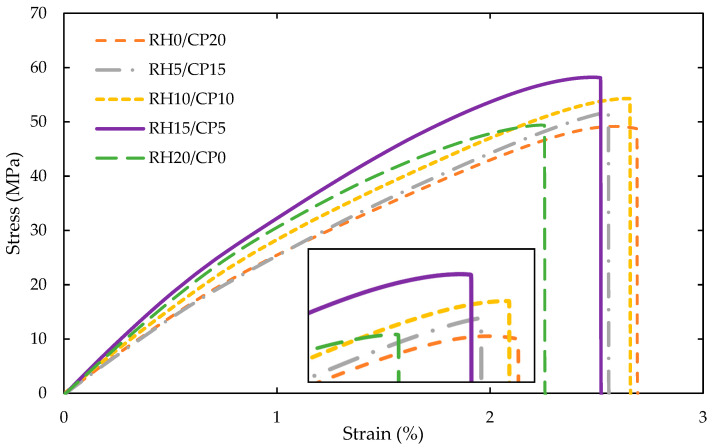
Typical stress-strain curves of RH/CP reinforced ABS blend composites.

**Figure 5 polymers-13-01171-f005:**
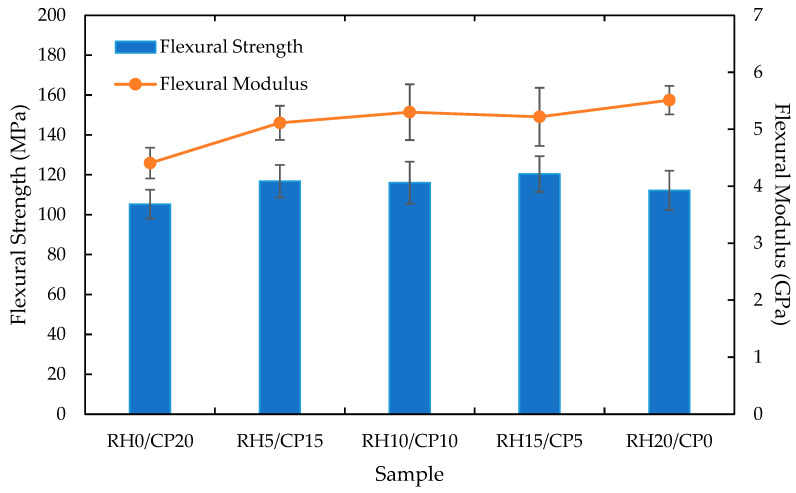
Flexural properties of RH/CP reinforced ABS blend composites.

**Figure 6 polymers-13-01171-f006:**
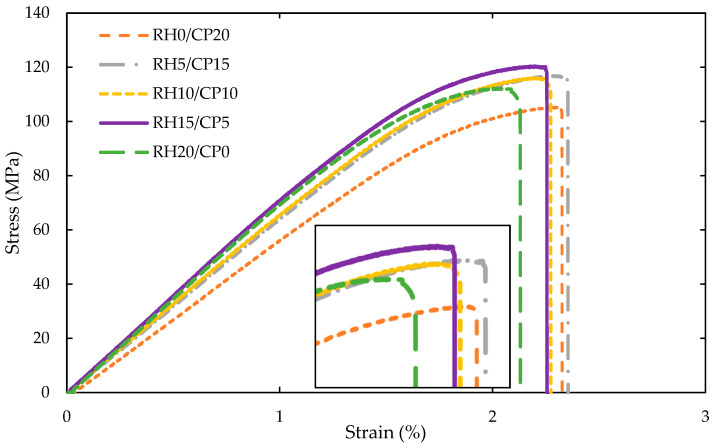
Typical flexural curves of RH/CP reinforced ABS blend composites.

**Figure 7 polymers-13-01171-f007:**
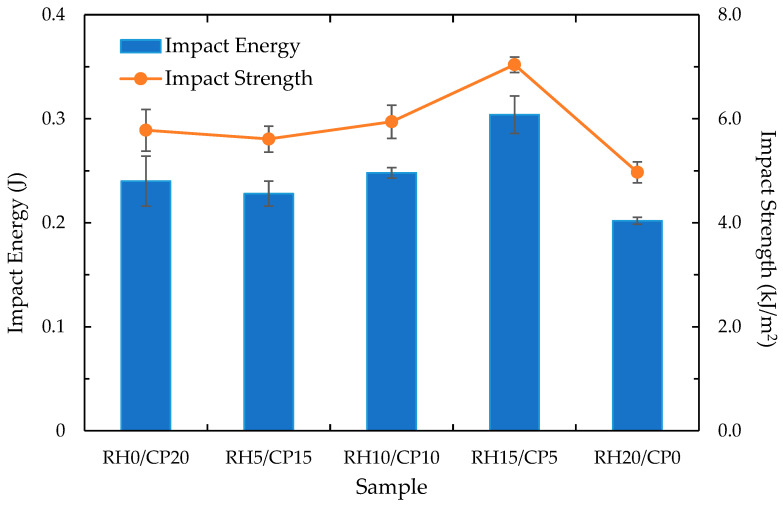
Impact behaviour of RH/CP reinforced ABS blend composites.

**Figure 8 polymers-13-01171-f008:**
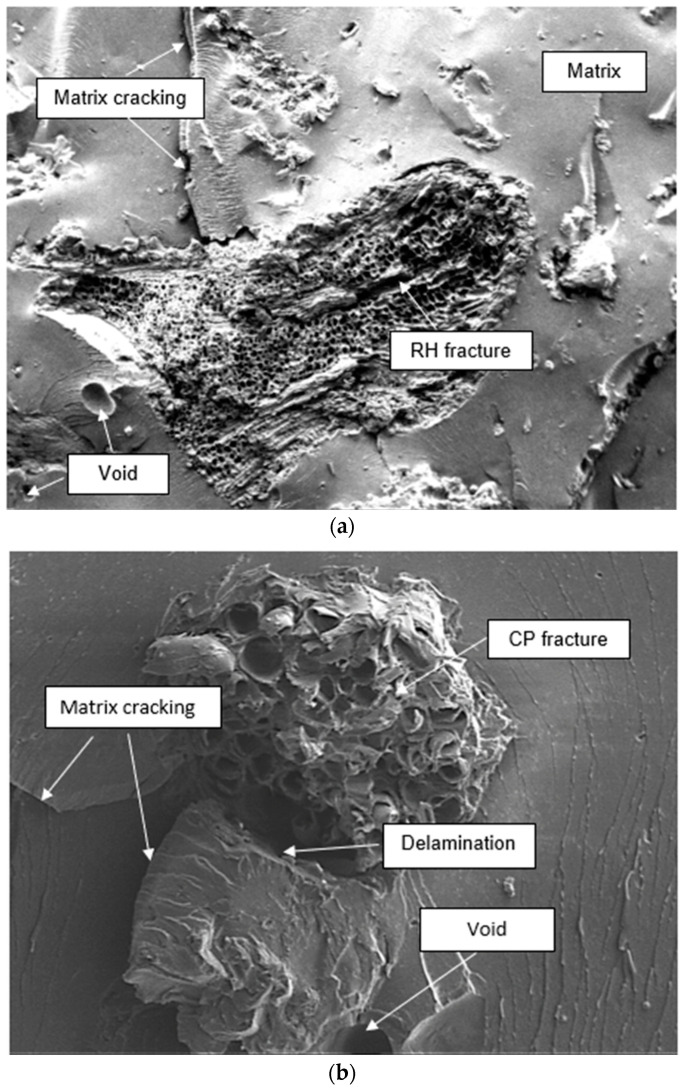
Scanning electron microscopy (SEM) micrographs of tensile fracture surface from (**a**) RH15/CP5 and (**b**) RH0/CP20 wt.%.

**Table 1 polymers-13-01171-t001:** Sample composition and designation.

Sample Composition (wt.%)	Designation
ABS (80) + Rice Husk (20) + Coco Peat (0)	RH20/CP0
ABS (80) + Rice Husk (15) + Coco Peat (5)	RH15/CP15
ABS (80) + Rice Husk (10) + Coco Peat (10)	RH10/CP10
ABS (80) + Rice Husk (5) + Coco Peat (15)	RH5/CP15
ABS (80) + Rice Husk (0) + Coco Peat (20)	RH0/CP20

**Table 2 polymers-13-01171-t002:** Saturation water absorption, water absorption constant, swelling exponent constant, and diffusion coefficient of RH/CP reinforced ABS composites.

Composition	*M*_∞_ (%)	*k*	*n*	*D* × 10^−6^ (mm^2^/s)
RH0/CP20	9.05	0.124	0.415	0.104
RH5/CP15	8.11	0.095	0.448	0.114
RH10/CP10	7.51	0.085	0.469	0.121
RH15/CP5	7.22	0.055	0.519	0.127
RH20/CP0	6.92	0.039	0.567	0.134

## Data Availability

The data presented in this study are available on request from the corresponding author.

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
