# Peer review of "Physicomechanical Properties of Rice Husk/Coco Peat Reinforced Acrylonitrile Butadiene Styrene Blend Composites"

_polymers, 2021, doi:10.3390/polym13071171_

Round 1
Reviewer 1 Report
This study reports on the synergistic effect of Rice Husk/Coco Peat Rein-2 on the Acrylonitrile Butadiene Styrene Hybrid Composites. And a lot of experimental results have been collected and discussion has been presented to support the claim that the Physicomechanical Properties of Hybrid Composites have been improved by the hybrid filler. In my opinion, it is a good job, and I would like to suggest to consider the following comments to improve it.
1. The quality of figure 1 is suggested to improve. And the density values are marked for easily catch.
2. The quality of figure 2 is suggested to improve. Figure 2b is not uniform to the others. Please put the log(t) down to the bottom.
3. The quality of figure 4 is suggested to improve. The maximum of each curve is suggested to mark for comparison.
4. The discussion on synergistic effect of Rice Husk/Coco Peat Rein-2 is not enough. Each filler has the inherent effect on the composite, however, there is no explanation for it. Why this hybrid filler has been chosen for composite, how about the other types of hybrid fillers? It is difficult to catch up with the claim.
5. For the effect of moisture, the discussion is not enough for the experimental results. As there is a diffusion of water molecules, the experimental properties are therefore depressed, there are so many previous works have been carried out for it. Please consider the following references to discuss the working principle behind the experimental results, to explore the working mechanism, i.e., (1) Huang, WM; Yang, B; An, L; Li, C; Chan, YS. Water-driven programmable polyurethane shape memory polymer: Demonstration and mechanism. APPLIED PHYSICS LETTERS. 2005, 86, 114105. (2) Haibao Lu, Yanju Liu, Jinsong Leng and Shanyi Du. Qualitative Separation of the Physical Swelling Effect on the Recovery Behavior of Shape Memory Polymer. European Polymer Journal. 2010, 46(9): 1908-1914. (3) Haibao Lu and Shanyi Du. A phenomenological thermodynamic model for the chemo-responsive shape memory effect in polymers based on Flory-Huggins solution theory. Polymer Chemistry. 2014, 5(4), 1155-1162.
In all, a useful study, and the working principle and theoretical analysis are suggest to strengthen.
Author Response
Response to Reviewer 1 Comments
Point 1: The quality of figure 1 is suggested to improve. And the density values are marked for easily catch.
Response 1: Figure 1 has been improved as suggested in line 183. (Yellow highlighted)
Point 2: The quality of figure 2 is suggested to improve. Figure 2b is not uniform to the others. Please put the log(t) down to the bottom.
Response 2: Figure 2 has been improved as suggested in line 217. (Yellow highlighted)
Point 3: The quality of figure 4 is suggested to improve. The maximum of each curve is suggested to mark for comparison.
Response 3: Figure 4 has been improved as suggested in line 289. (Yellow highlighted)
Point 4: The discussion on synergistic effect of Rice Husk/Coco Peat Rein-2 is not enough. Each filler has the inherent effect on the composite, however, there is no explanation for it. Why this hybrid filler has been chosen for composite, how about the other types of hybrid fillers? It is difficult to catch up with the claim.
Response 4: The comments are accepted and the additional discussions on synergistic effect had been added in line 88-93 and line 370-373. (Yellow highlighted)
Point 5: For the effect of moisture, the discussion is not enough for the experimental results. As there is a diffusion of water molecules, the experimental properties are therefore depressed, there are so many previous works have been carried out for it. Please consider the following references to discuss the working principle behind the experimental results, to explore the working mechanism, i.e., (1) Huang, WM; Yang, B; An, L; Li, C; Chan, YS. Water-driven programmable polyurethane shape memory polymer: Demonstration and mechanism. APPLIED PHYSICS LETTERS. 2005, 86, 114105. (2) Haibao Lu, Yanju Liu, Jinsong Leng and Shanyi Du. Qualitative Separation of the Physical Swelling Effect on the Recovery Behavior of Shape Memory Polymer. European Polymer Journal. 2010, 46(9): 1908-1914. (3) Haibao Lu and Shanyi Du. A phenomenological thermodynamic model for the chemo-responsive shape memory effect in polymers based on Flory-Huggins solution theory. Polymer Chemistry. 2014, 5(4), 1155-1162.
Response 5: The paper conducted the simple weight differential water absorption study on the hybrid composites and further discussed with Fickian’s theory to obtain the hybrid composites’ diffusion coefficient. Thus, the authors think that the discussion about the depressed experimental properties because of water molecule’s diffusion is not necessary to the study. However, the suggested papers are very good to be included as the references in the study. Therefore, the papers had been added as the references at number 24 and 25. (Yellow highlighted)
Additional Response:
- The article had been proofread and grammar checked by the professionals.
- I would like to pass on the appreciation and thank you to reviewer 1 for the supportive comments.

Reviewer 2 Report
See the comments to the Editor.
Author Response
The article had been proofread and grammar checked by the professionals. I would like to pass on the appreciation and thank you to the reviewer 2 for the supportive comments.
Round 2
Reviewer 2 Report
See the comments to the Editor.
Author Response
Point 1: Authors did not answer all these concerns or questions.
Response 1: Sorry for leaving out your comments as the author cannot see the comments previously.
Point 2: Authors said" hybrid", however, it cannot be concluded from this work. "Hybrid" is unprecise here. Delete it or replace it by "blend".
Response 2: The comment is accepted, and the word “hybrid” had been replaced by “blend” as suggested. (Green highlighted)
Point 3: Materials detailed information should be provided for ABS, RH, and CP.
Response 3: A more detailed information is added on ABS, RH, and CP as suggested in line 106-107, 108-109, and 109-110 respectively. The more detailed explanations on the processes were already added in the next paragraph. (Green highlighted)
Point 4: The color curves for Figures 2, 3, and 5 are better and clearer than black ones.
Response 4: The color had been added to Figure 2, 3, 5 and 7 as suggested. (Green highlighted)
Additional Response:
- The article had been proofread and grammar checked by the professionals.
- I would like to pass on the appreciation and thank you to reviewer 1 for the supportive comments.
